# Language Models Can Predict Their Own Behavior

**Dhananjay Ashok**
Information Sciences Institute
University of Southern California
ashokd@usc.edu

**Jonathan May**
Information Sciences Institute
University of Southern California
jonmay@isi.edu

## Abstract

The text produced by language models (LMs) can exhibit specific 'behaviors,' such as a failure to follow alignment training, that we hope to detect and react to during deployment. Identifying these behaviors can often only be done *post facto*, i.e., after the entire text of the output has been generated. We provide evidence that there are times when we can predict how an LM will behave early in computation, before even a single token is generated. We show that probes trained on the internal representation of input tokens alone can predict a wide range of eventual behaviors over the entire output sequence. Using methods from conformal prediction, we provide provable bounds on the estimation error of our probes, creating precise early warning systems for these behaviors. The conformal probes can identify instances that will trigger alignment failures (jailbreaking) and instruction-following failures, without requiring a single token to be generated. An early warning system built on the probes reduces jailbreaking by 91%. Our probes also show promise in pre-emptively estimating how confident the model will be in its response, a behavior that cannot be detected using the output text alone. Conformal probes can preemptively estimate the final prediction of an LM that uses Chain-of-Thought (CoT) prompting, hence accelerating inference. When applied to an LM that uses CoT to perform text classification, the probes drastically reduce inference costs (65% on average across 27 datasets), with negligible accuracy loss. Encouragingly, probes generalize to unseen datasets and perform better on larger models, suggesting applicability to the largest of models in real-world settings. [1]

## 1 Introduction

Language models (LMs) have emerged as the dominant approach to general language tasks [88, 22], seeing widespread adoption in chatbots [3, 68], code generation [18] and reasoning systems [67]. However, they have been known to 'misbehave,' hallucinating false information [98], failing to adhere to output specifications [59], or, most concerningly, producing content that is misaligned with human values [75], sometimes dangerously so [35]. In practice, such misbehavior is mitigated by deploying *post-hoc* guardrails on the output of the LM [92, 24, 16, 101, 8, 53]. For example, a chatbot LM may exhibit harmful behavior by complying with a malicious request [41, 90]; developers will typically attempt to detect this behavior in the output [101] and return an abstention message instead of the original LM content. Such a framework suffers the economic [100] and environmental [83, 76] expense of generating every output token, a cost that threatens to grow exponentially as frontier labs continue to scale both model sizes [37, 49] and the number of tokens generated during inference [91, 95, 67, 32].

In this paper, we show that the internal representations of LMs can be used to **preemptively** predict behaviors that will emerge over the entire output sequence. This information becomes accessible

---

[1] Our code is accessible at: https://github.com/DhananjayAshok/LMBehaviorEstimation

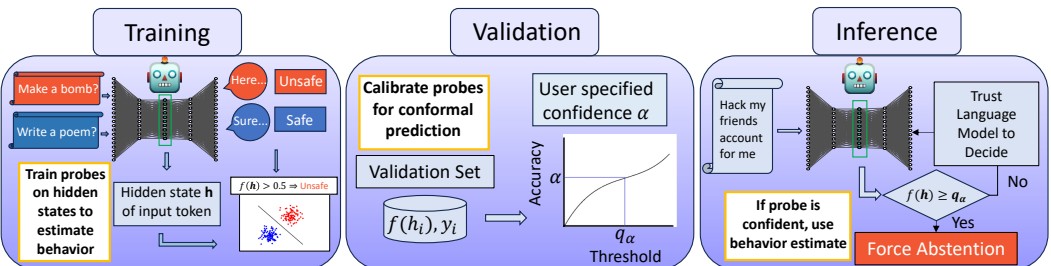

Figure 1: Overview of our method as applied in preemptive safety alignment failure detection. During training, we use many input prompts to collect a dataset consisting of the hidden states of the input tokens and eventual output behavior (in this example, the behavior is whether the LM fails to abstain on a malicious prompt). The dataset is used to train a probe on the hidden states. With a validation set, we calibrate the probes for conformal prediction by identifying a threshold for confident predictions. During inference, if the probe is confident in its estimate of the eventual LM behavior, we stop computation early (in this example, overriding the LM and abstaining).

**before the LM has generated a single token**, enabling the system to take appropriate actions without suffering inference costs on those instances. Concretely (Figure 1), we train linear classifiers (probes) [4] that use an LM's internal representation of input tokens to predict the eventual behavior of its output. We then calibrate the probes using methods from conformal prediction [81]. During inference, we look to probes to make a prediction only when there is a provable guarantee on the estimation error, ensuring precise early warning signals for various model behaviors.

We show that these probes can preemptively identify degenerate behavior, such as instruction-following failures (Section 3). Across multiple datasets and output formats, our probes give precise early warning signals for cases where the LM will fail to follow specified output format instructions.

Conformal probes can also detect deeper, 'self-reflective' properties, like whether an LM's behavior conflicts with its safety alignment. We demonstrate this (Section 4.1) by applying our method to enhance the safety of an LM that is instructed to abstain from complying with malicious requests. Probes are trained to identify cases where this will not happen—the request will be malicious **and** the model will not abstain. Once again, our probes are able to precisely detect (with over 92% accuracy) cases where the LM complies with malicious requests. An early warning system built on these conformal probes reduces the jailbreak success rate from 30% to 2.7%, a 91% reduction.

We further show that conformal probes can identify behaviors that cannot be measured with output text. We show (Section 4.2) that our probes can *a priori* estimate an LM's **confidence** as measured by the per-token perplexity of the output; as this model-specific property cannot be inferred from just the text sequence alone, we show the wide scope of applicability of our probing approach.

Finally, we apply our method to an LM that tackles text classification by generating several explanation tokens before a class prediction, i.e. using Chain-of-Thought (CoT) reasoning [91] (Section 5). The probes estimate the final class prediction that will appear after the reasoning chain and exit early if confident in their estimate. On 27 datasets, spanning the tasks of Multiple Choice QA, Sentiment Analysis, Topic Classification, Toxicity Detection and Fact Verification, our method reduces inference costs by 65% on average, while suffering accuracy losses of no more than 1.4% (worst case). The probes generalize to unseen datasets, reducing the inference cost of CoT on OpenBookQA [63] by 68% with minimal loss to accuracy (0.4%), despite only training on other MCQA datasets.

We demonstrate that often fewer than 500 training instances are sufficient for the probes to attain high estimation consistency, while ablations on the probing layer present a more complicated, task-specific story. Encouragingly, increasing the scale of the LM improves the performance of our technique, suggesting it may scale favorably with ever-increasing model sizes [15, 37, 19, 26].

We hope our findings enable practitioners to build efficient early warning systems on language models and enable further research into the information encoded in their hidden states [71, 10, 66].

## 2   Background

**Hidden State Probing:** Lightweight probes have long been used to interpret the internal activations of neural networks [4] and language models [71, 10]. Given a set of input prompts, we compute the generated outputs and store the internal activations computed during the forward passes of an LM (e.g. the outputs of a specific Transformer [88] layer). The instances are then assigned a classification label meant to capture some property of the input and output. The probes are then trained to predict the label using only the internal activations as input. For example, Azaria and Mitchell [10] collect the activations of the middle layer of an LM when processing the output tokens, and manually label the generations as true or false. They then train a small neural network which can identify whether the model output is truthful, given the internal activations of the generated tokens. Prior work in probing uses the internal states of all tokens, even generated ones, which incurs the cost of inference. In this work, we explore an alternate approach, training probes on the internal states of the input tokens alone, allowing us to predict properties of the output tokens **before** they have been generated.

**Conformal Prediction:** Given a validation set, conformal prediction can be used to precisely calibrate the model's confidence [29, 52]. During inference, the conformal system makes a prediction if and only if the probability of the prediction being accurate is above a user-specified probability [81].

In the classification setting, we are given a validation set $\{(\mathbf{x_1}, \ldots \mathbf{x_n}), (y_1, \ldots y_n)\}, \mathbf{x}_i \in \mathbb{R}^d, y_i \in \{1, 2, \ldots, c\}$ (where $d$ is the dimension of the input vector and $c$ is the number of classes), a classifier which maps the input to scores for each class $f : \mathbb{R}^d \to \mathbb{R}^c$ and a user-defined confidence level $\alpha \in (0, 1)$. To form a well-calibrated prediction set, we define a score function $S : \mathbb{R}^d \times \{1, 2, \ldots c\} \to \mathbb{R}$ that measures how poor a prediction is e.g. $S(\mathbf{x}, y) = 1 - (\frac{\exp f(\mathbf{x})}{\sum_i \exp f(\mathbf{x})_i})_y$. Using scores on the validation set $(s_1, \ldots s_n)$ we calibrate by calculating threshold $\hat{q}_\alpha$, the $1 - \alpha$ quantile of the scores:

$$\hat{q}_\alpha = \text{Quantile}(\{s_1, s_2, \ldots s_n\}, \frac{\lceil (n + 1)(1 - \alpha) \rceil}{n}) \tag{1}$$

At inference time, the prediction set for a test instance $\mathbf{x}$ is $\{\hat{c}|S(\mathbf{x}, \hat{c}) \le q_\alpha\}$. For the above score function, no two classes can score greater than 0.5 simultaneously, implying that with $\alpha > 0.5$ the prediction set at inference time is either empty (deferral/abstention) or consists of a single class label.

If the validation set is *exchangeable* (a slightly weaker assumption than the typical I.I.D assumption [12]) with the test set, then a prediction at inference time is provably guaranteed [81] to satisfy

$$\mathbb{P}[y_{\text{test}} = \hat{c}_{\text{test}}] \ge 1 - \alpha \tag{2}$$

## 3   Can Internal States Predict Eventual Behavior?

In this section, we ask whether the internal representations of the LMs input can be used to predict eventual behavior, even before any output token has been generated. We explain our methodology with the running example of an LM instructed to answer a question in bullet point format.

The LM is expected to provide an answer in two bullet points, with any fewer or greater bullets in the answer being a failure to follow instructions. Suppose we are given an input prompt:

```
Question: What do Osteoclasts do?
```

A LM may respond to this with:

```
1. Decompose bone 2. Differentiate from monocyte 3. Create space for tissue
```

We can identify that the LM has behaved contrary to its instructions when it generates the '3'. The previous tokens give no indication that a failure was imminent; however, is it possible that the model contains signals that can identify this eventual failure by the time it generates the first token, i.e. '1'?

To test this, we collect three QA datasets—NaturalQA [54], MSMarco [65] and TriviaQA [47] and prompt the LM to answer questions in a specific format. One format requires the answer to be organized in exactly three bullet points, while the other requires a JSON output with pre-specified

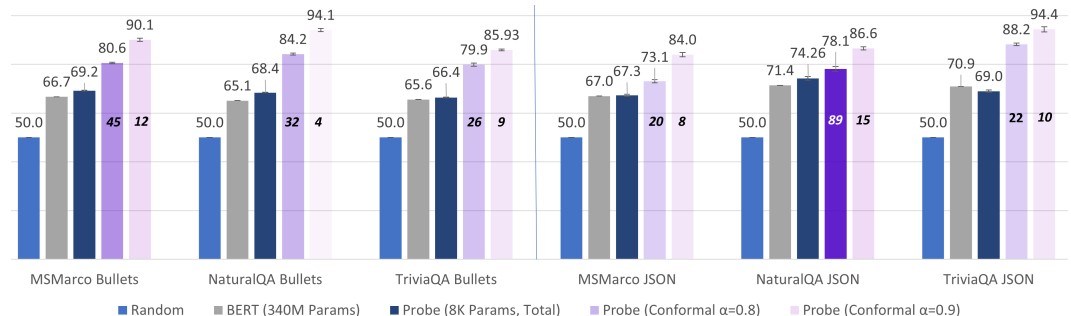

Figure 2: Estimation consistency, i.e. accuracy when preemptively predicting whether the LM will fail to follow format instructions. Numbers inside the bars in ***bold italics*** are the coverage percentages for conformal probes that engage in selective prediction. All probe results are means over 5 random seeds, with error bars showing the $2\sigma$ confidence interval. The probes not only outperform the random baseline, but consistently match or outperform fine-tuning on a BERT model despite using fewer than 0.003% of the parameters. The conformal probes have significantly higher consistency, and give users the flexibility to trade off precision and coverage based on the confidence level $\alpha$.

fields. We evaluate whether the output format has been followed, and then sample this data to obtain training and testing splits that have an equal number of failures and successes.

We collect the output of the middle Transformer layer of Llama3.1-8B [26] when processing the final token of the input (in the example above, the '?'), then train linear classifiers to predict whether the output will **eventually** fail to follow instructions. We ablate layers in Section 6, reproduce results on other LMs in Appendix A, and provide the prompts that specify the format in Appendix C.

We also fine-tune a bert-large-uncased [22] model for text classification, which involves tuning 340M parameters, $4 \times 10^4$ times the number of trainable parameters of the linear model. The BERT model is trained to predict whether the output will fail to follow formatting instructions given the **text of the input prompt alone**, and is a measure of how a significantly stronger model performs when attempting to establish a pattern between the inputs to the model and the output behavior. Methods are evaluated on the metric of *estimation consistency*, i.e, how accurate they are at preemptively predicting whether or not the formatting instructions will be followed.

The probes (Figure 2, navy blue bar) outperform the random baseline in all settings, showing that an LM's representation of the input tokens contains information on behavior that will emerge over the entire output sequence. Moreover, the probes also outperform fine-tuning on BERT in all but one of the settings, despite using around only 0.0025% as many parameters.

However, despite outperforming the baselines, the average estimation consistency of the probes is less than 75%. A naive attempt to act on the probes estimate would result in catastrophic failure.

## 3.1 Creating robust behavior estimators with conformal prediction

Internal states may occasionally be insufficient to estimate eventual behavior. The ideal system handles such cases, making consistent predictions when confident and deferring otherwise. Hoping to impart such a capability to our probes, we use the probes learned above in a conformal prediction framework. Specifically, we use a held-out validation set $\mathbb{D}_{\text{valid}}$ to calibrate the probe after training. We compute the probes' prediction probabilities $\hat{\mathbf{y}} \in [0, 1]^{|c|}$ for each class with true label $y \in \{1, \ldots c\}$ on the validation set, and find the lowest threshold quantile $q$ that satisfies:

$$\frac{\sum_{\mathbf{y_i} \in \mathbb{D}_{\text{val}}} \mathbb{I}[(\max(\hat{\mathbf{y}_i}) \geq q) \wedge (\text{argmax}(\hat{\mathbf{y}_i}) = \mathbf{y_i})]}{\sum_{\mathbf{y_i} \in \mathbb{D}_{\text{valid}}} \mathbb{I}[\max(\hat{\mathbf{y}_i}) \geq q]} \geq \alpha \qquad (3)$$

During inference, when a probe predicts a behavior with probability vector $\hat{\mathbf{y}}_{\text{test}}$, we return the prediction if and only if $\max(\hat{\mathbf{y}}_{\text{test}}) \geq q$, otherwise we defer to the LM. Unless specified otherwise, we set $\alpha = 0.9$. If no satisfying $q$ can be found, we defer on all instances. We ablate $\alpha$ in Section 6.

| Dataset | Conformal Probe ($\alpha = 0.9$) | | | | BERT | | |
| --- | --- | --- | --- | --- | --- | --- | --- |
| | Coverage | Consistency | Prec. | Rec. | Consistency | Prec. | Rec. |
| SelfAware | 100.0 | **98.0** | **94.5** | **94.5** | 93.3 | 91.7 | 84.4 |
| KnownUnkown | 92.3 | **92.6** | **75.1** | 72.8 | 67.1 | 64.4 | **93.1** |
| WildJailbreak | 100.0 | **94.0** | **90.0** | **91.0** | 83.4 | 73.8 | 33.3 |

Table 1: Probes estimate whether the LM will fail to abstain when it should have with significantly higher precision and recall than a BERT baseline. On WildJailbreak, where 30% of malicious prompts are mistakenly complied with, using the probe as an early warning signal reduces successful jailbreaks to 2.7%. In comparison, using the BERT model would leave jailbreak success rate at 20%.

Using conformal probes (Figure 2) significantly increases performance across all datasets. At $\alpha = 0.8$, the probes have a test coverage of 40% and, for the covered datapoints, estimation consistency is 80.7%, showing that the conformal probes are well calibrated to the user-specified confidence level. Raising $\alpha$ to 0.9 results in a reduced average coverage of 10%, but an increased average consistency of 89.2%. This allows users to flexibly trade off between coverage and consistency, enabling precise and trustworthy early warning systems that do not offer an opinion when confidence is low.

## 4 Creating Early Warning Systems for Other Behaviors of Interest

In this section, we demonstrate that probes are not limited to preemptively identifying degenerate behavior, such as failing to follow instructions. We show that conformal probes can serve as early warning systems for instances that will trigger failures to follow safety alignment, and those that will result in the model giving a low-confidence response. We describe the key experiment designs below, with exact prompts and details in the Appendix C.

### 4.1 Detecting failures to follow safety alignment

The deployment of LM-powered chatbots raises concerns of harmful behavior that is unaligned with human interests [28]. One common setting where such behaviors arise is when the user requests information that does not exist, with models often hallucinating an answer regardless, misleading the user in the process [9]. Another common occurrence is when the user themselves requests the model to help them complete malicious tasks, such as building a weapon of mass destruction [90].

We collect two datasets (SelfAware [96], KnownUnknown [5]), which have answerable and unanswerable questions. The LM reasons as to whether the question is answerable, providing a response if it is and abstaining otherwise. Similarly, we collect malicious and benign prompts from WildJailbreak [45], and prompt the LM to refuse to comply with a request if it is malicious. Inspired by work in AI safety and alignment, we adopt the viewpoint that answering an unanswerable question or complying with a malicious request is disproportionately problematic [11]. We filter our training set to only include the instances where the LM answers or complies with the request, and train probes to identify the cases where the LM should have abstained (i.e. fails to avoid answering an unanswerable question or falls victim to jailbreaking). During inference, if the probe is confident ($\alpha = 0.9$) that the LM has failed to abstain when it should have, we override the LM and abstain instead.

Results show (Table 1) that probes greatly enhance the safety of the LMs, outperforming the BERT baseline on consistency, precision and recall. Across all datasets, probes identify over 72% of the failures to abstain (over 90% for 2/3 datasets). This safety comes at little cost; when the probes force an abstention, they are correct over 86% of the time (on average). They make confident estimations on at least 92% of the samples, and maintain conformal estimation consistency of at least 92%. Concretely, on WildJailbreak the use of conformal probes reduces the percentage of successful jailbreaks from 30% to 2.7%, a significant safety improvement.

### 4.2 Quantifying LM uncertainty

Language models are prone to hallucination [98] and often have gaps in their knowledge [38], making methods that estimate the trustworthiness of specific outputs vital. Recent work [48, 46] approaches this problem by using the model itself to quantify how certain it is in its answer, either by explicitly

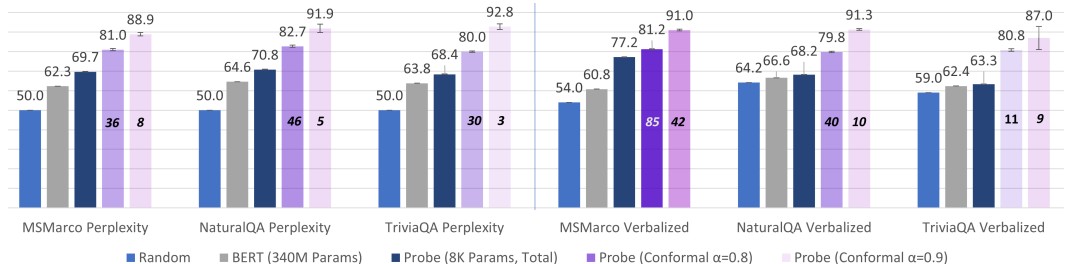

Figure 3: Estimation consistency when preemptively predicting an LM's confidence in its response. Numbers inside the bars in ***bold italics*** are the coverage for conformal probes that engage in selective prediction. Probes outperform all other methods, with the confidence estimation tasks proving more challenging for BERT. Conformal probes have lower coverage rates than the format following (Section 3) and safety alignment (Section 4.1) tasks. However, they maintain consistency at the user-defined confidence level, ensuring that their early warning signals are reliable.

prompting it to verbalize its confidence or using the per-token perplexity of the output. If we detect an untrustworthy response, we may abstain from answering or escalate the query to a more capable model. An early warning system that preemptively detects such responses before they are even generated would allow us to take these actions more efficiently.

We train conformal probes to preemptively estimate how confident the LM will be in its response using both of these uncertainty quantification methods on NaturalQA, MSMarco and TriviaQA. In the case of the perplexity measure, we consider the bottom $25\%$ of scores to be 'high confidence,' and the top $25\%$ to be 'low confidence' (discarding the rest). In both cases, the probes attempt to identify whether the model will be confident in its output. This task proves more challenging for all methods, however, probes continue to outperform both the random baseline and the much larger BERT model. Despite this, the conformal probes maintain high consistency, making the early warning and redirection system feasible in practice.

## 5  Selective Inference-Scaling with Conformal Probes

In this section, we show how our method can be used to accelerate systems that use CoT prompting with an LM to perform text classification. We collect 27 text classification datasets, spanning the tasks of Multiple Choice Question Answering, Sentiment Analysis, Topic Analysis, Toxicity Detection and Fact Verification. Details on dataset setup are in Appendix C.

We use Llama3.1-8B under CoT prompting to perform text classification, i.e., outputting an explanation before the final class prediction. We train a linear probe that uses the internal representation of the final input token at the 18th layer (based on results from previous experiments) to predict the class that the CoT model will eventually output, and then perform conformal calibration. During inference, if the conformal probe is confident, we interrupt generation and use the probe estimation as the final answer. If not, we allow the model to continue its CoT generation and provide the final answer. We note that this method of acceleration is **synergistic** with other inference optimization algorithms such as Speculative Decoding [55] or architectural changes [27]. This is because in the cases where the probe does not interrupt the LM, it does not influence the remaining forward passes, hence conformal probes can be flexibly added on top of any inference paradigm for further efficiency.

We compare to vanilla CoT on two metrics—Accuracy Loss (Accuracy of CoT - Accuracy of Method) and Inference Cost Reduction ($\frac{\text{\#CoT Forward Passes - \#Method Forward Passes}}{\text{\# CoT Forward Passes}}$).

The results (Figure 4, in blue) show that the method is highly effective at reducing the inference cost with minimal cost to accuracy. The minimum inference cost reduction is $4.7\%$, with an average reduction of $65\%$ across all datasets. Despite this significant speedup, the average accuracy loss is near zero ($-\mathbf{0.46\%}$), with the worst loss at $1.34\%$. Surprisingly, accuracy **increases** on several datasets. Finally, we investigate the out-of-distribution generalization capabilities of the conformal probes. For each test dataset of the MCQ and Sentiment tasks, we train and calibrate using data from every **other** dataset. The results (Figure 4, in orange) show that the probes do exhibit OOD

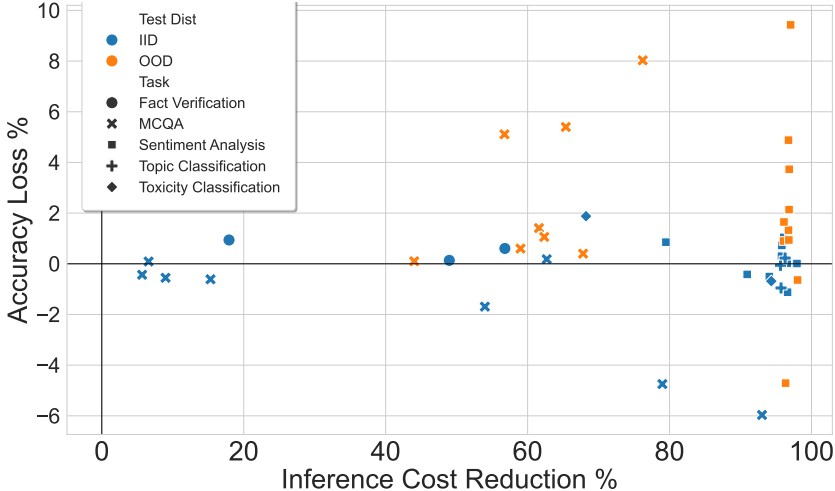

Figure 4: Tradeoff between accuracy loss and inference cost when using probes to accelerate CoT Prompting in IID and OOD settings. Using IID probes leads to a 1.34% accuracy loss (worst case) and a **negative** accuracy loss on average (i.e. accuracy improves). Inference cost reductions are 65% on average. Probes generalize to OOD data, inference cost reductions are higher at 81% on average, with a small accuracy loss of 2.3% on average.

generalization, suggesting the method may be applicable even when there are slight shifts between training and test distributions.

## 6   Analysis and Ablations

**Why does accuracy improve when using probes?** Seeking to explain the surprising increase in accuracy when accelerating CoT models (Figure 4), we plot (Figure 5, left) the correlation between the CoT model's accuracy and the probe's estimation consistency. It is generally positive, which suggests that incorrect *probe* estimation correlates with incorrect CoT generation. This minimizes the harm of an inconsistent estimation of the CoT decision, as the cases where estimates are likely to be inconsistent coincide with cases where a consistent estimate does not benefit accuracy.

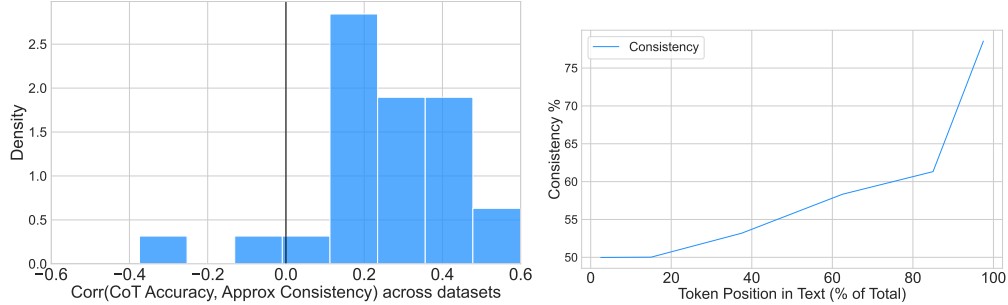

Figure 5: Left: Density plot of the correlation coefficient between the CoT LM accuracy and the consistency of the probe. The coefficient is often positive, indicating that instances where the probe inaccurately estimates the LM decision are also instances where the LM is more likely to be incorrect. Right: On CosmoQA, probes that use the initial tokens of the input prompt have near-random consistency, with performance increasing considerably as we use later tokens in the prompt.

**How early in the computation does the model show signs of the final behavior?** We have shown that the final input token often contains sufficient information to predict output behavior, but at which token does this information **start** becoming clear and accessible? To investigate this, we collect activations from **every** input token (after the FewShot example tokens) of the CSQA dataset and train

a linear classifier to map these internal activations to the eventual CoT prediction. Probing (Figure 5, right) using the embeddings of tokens in the first quarter of the question leads to near-random performance, showing that this information is not readily accessible at initial tokens. Consistency increases steadily as we use later tokens, with a sharp rise around the final tokens.

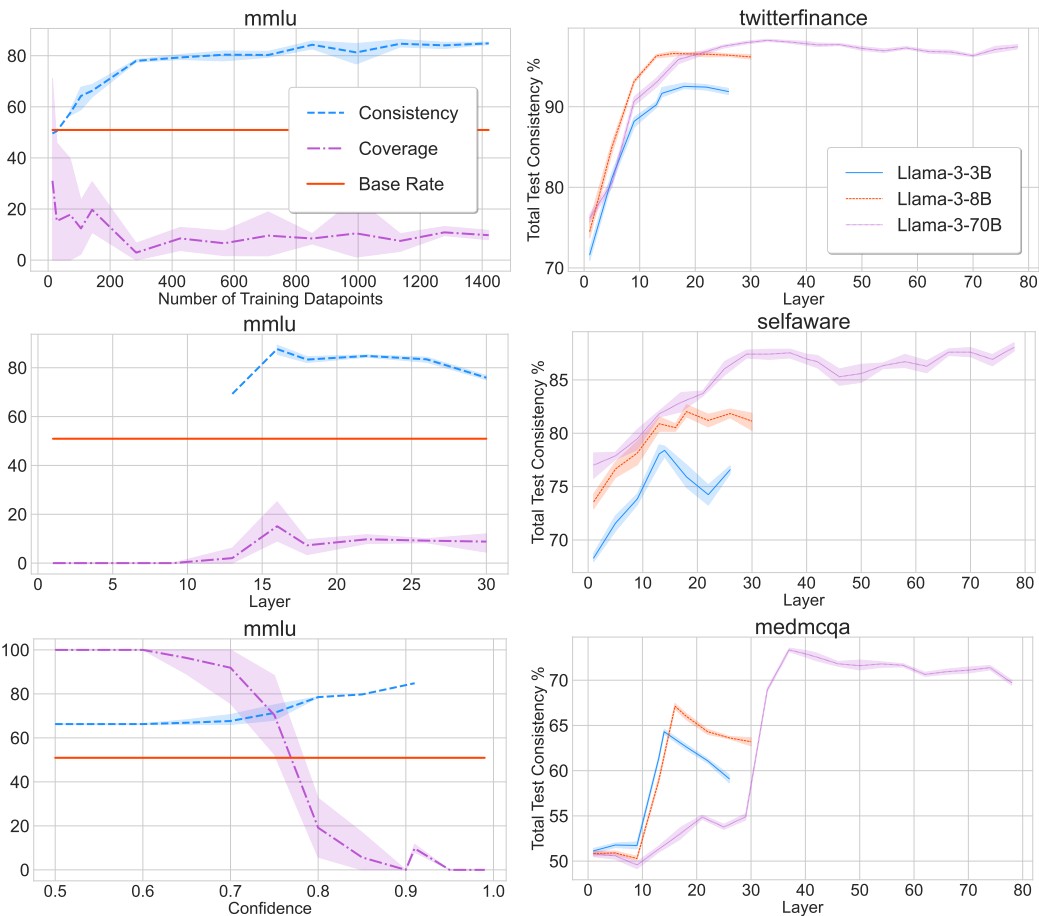

Figure 6: **Left**: Ablations on MMLU. Within 500 training datapoints, probes ($\alpha = 0.9$) often approach the consistency and coverage they will attain when trained on the entire dataset. Layer ablations ($\alpha = 0.9$) show that early layers perform poorly. Increasing the confidence threshold $\alpha$ has no effect until a point, after which conformal consistency increases while the coverage tends to zero. **Right:** Estimation consistency increases with LM size, suggesting the method may scale favorably

**Ablations on probe parameters:** We ablate the parameters involved in the design of the conformal probing system: number of training datapoints, probing layer and the conformal confidence threshold $\alpha$. We show results (Figure 6, left) for the MMLU dataset, however, all trends discussed hold for the majority of the datasets we explored, with more examples in Appendix B.

We ablate the amount of data used to train the probe and find (Figure 6, top left) that probes can approach the consistency and coverage they will achieve over the entire dataset with only 500 datapoints, suggesting that the method is particularly data efficient. We explore (Figure 6, middle left) whether the specific layer being probed affects the performance of the probe. Our most general finding is that early layers offer poor estimation consistency. However, whether mid or late layers perform better is task and dataset-specific, and cannot be described generally. For more details, see Appendix B. Increasing the confidence required from the probes has predictable effects (Figure 6, bottom left). Increasing the confidence threshold has no effect until a point, after which conformal consistency increases while the coverage tends to zero.

**How does model scale affect performance?** We varied the size of the LM used (Llama3 3B, 8B and 70B) and measured probe consistency across several layers. Encouragingly (Figure 6, right), the

performance of the larger models is consistently better, suggesting that the information encoded in the internal activations is easier for the probe to 'read' when the model is more powerful. This bodes well for the methods' ability to scale with models of increasing size and capability.

**What are some limitations of the probes?** Other experiments we conducted suggest that probes are limited in the behaviors they can detect. Using the MCQ task, we tried to estimate whether the LM would output an **incorrect** answer. The probe accuracies were consistently near random. We hypothesize that since probes are simple linear classifiers, they can only detect patterns and use 'knowledge' that is well encoded in the LM activations. This suggests that probes struggle with output properties that cannot be identified by the LM itself (without external knowledge). We also observed that the correlation between the estimation consistency of the probe and the token count of the output is negative (-6.1%). This suggests probes struggle more with inputs that evoke longer outputs.

# 7    Related Works

This work takes inspiration from recent methods that probe the hidden states of LMs to observe interpretable patterns [4, 51, 71, 36, 3] identify false statements [10, 56, 57, 97] and hallucinations [20, 84, 44]. These works use the internal states of every token, even the outputs, to produce signals for these phenomena. In contrast, our work shows that internal states can predict behaviors **before any** output tokens are generated, suggesting that input token embeddings contain rich information on future LM behavior. Furthermore, while conformal prediction has been used with LMs to guarantee the quality of generated text [73], robotic trajectories [89, 74] and text classification [52, 17], we are the first to use it in conjunction with hidden state probes to create precise early warning systems.

Our work also has connections to the literature on early exiting during the forward pass of NN models, with works often using signals from the hidden states to prematurely exit with a prediction on the future outputs [93, 99, 94, 80, 42, 70]. While these works focus on the tokens predicted, we show a more general result: that internal states can predict the future behavior of LMs. These include behaviors such as whether or not the LM will output a high perplexity answer, or mistakenly comply with a malicious request, which cannot be inferred just by observing the output token sequence.

While recent work has begun exploring whether LMs exhibit 'introspective' properties in the black box setting [13], or whether LMs can predict 'global attributes' of their response [72, 25], we are the first to show that when internal probes are deployed under conformal prediction they can be used to create early warning systems for a wide range of behaviors like question abstention to format following errors. Our work advances research on understanding the nature of the information contained in the hidden states of LMs [71, 6, 66, 58, 87, 62]. Specifically, we show that the information contained in the hidden states is relevant not just to the next token, but to behaviors that manifest several tokens later during the LMs generation.

# 8    Conclusion

We show that a language model's hidden representation of input tokens alone contains vital information on the behavior of the LM over the entire output sequence. We train linear probes to read this information and use it in a conformal prediction framework to create precise early warning or exit systems for a wide range of LM behaviors, including degenerate behaviors, safety alignment failures and more. The conformal probes can preemptively identify, before a single token is generated, instances where an LM will fail to abstain from answering an unanswerable question, fall victim to jailbreaking, fail to follow output format specifications or give low-confidence responses. On 27 text classification datasets across 5 different tasks, the method can accelerate Chain-of-Thought prompting by 65% with little accuracy loss. We show that the probes generalize to out-of-distribution test sets and scale favorably to larger LMs. Finally, we explore the limitations of the method, showing that the behavior of longer output sequences is harder to estimate and that tasks that require knowledge external to the model are particularly challenging. With the rising popularity of inference-time scaling methods, we hope our work can help ameliorate the growing computational cost of running LMs and provide more insight into the nature of the information contained in their hidden states.

## Acknowledgments

We thank Robin Jia for his exceptional course on The Science of Large Language Models (accessible at `https://robinjia.github.io/classes/fall2024-csci699.html`) and his advice on the early stages of this project. We also acknowledge support from Open Philanthropy.

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

# A Confirming Robustness of Results

## A.1 Confirming Robustness to Choice of Model

We conduct experiments on two other language models to ensure that our results hold for language model families outside of the Llama3 series. We use Mistral-7B-Instruct-v0.3 from MistralAI [43] and DeepSeek-R1-Distill-Qwen-14B from DeepSeek [32]. The results show that the phenomenon shown in the main text can be detected robustly across various model families.

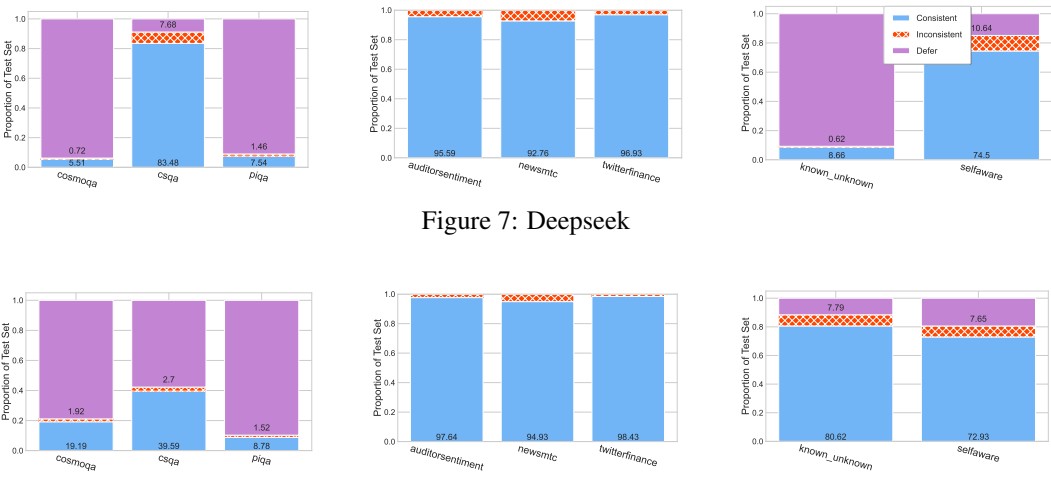

Figure 7: Deepseek

Figure 8: Mistral

## A.2 Data Leakage

Several datasets used in this study have been released before the creation of Llama3, and as such, might have been pre-exposed to the model during training. We detail our reasons as to why data leakage is not a concern for the conclusions that we draw from this study:

- There are datasets in our study that have not been leaked. For example, the MMLU test set is used as an official test benchmark by LLama3[26], and WildJailbreak was released to the public only after the initial release of Llama3 models [45]. The results are strong on such datasets as well, showing that data leakage is not behind the performance of the probes.
- Several tasks require the probes to identify behavior that fundamentally does not exist in text form on the internet. The abstention failure, format following (bullets and JSON) and confidence estimation tasks require the probes to determine how the model will behave, regardless of whether the model has seen the input text or not. For example, whether or not the model was pretrained on MSMarco questions is immaterial for predicting whether or not it will output answers in a specific JSON format.

# B Extended Analysis

## B.1 Training Datapoint Ablation

As a general trend (Figure 9), 500 training datapoints are usually sufficient for probes to achieve the maximum consistency and coverage that they will attain when trained on the entire dataset.

## B.2 Layer Ablation

A deeper analysis of the interaction between the performance and the layer used for probing shows that no consistent patterns can be identified. A subset of the common patterns (Figure 10) shows that sometimes that apart from early layers being poor, it is unclear whether the middle or final layers are consistently better for accuracy or confidence.

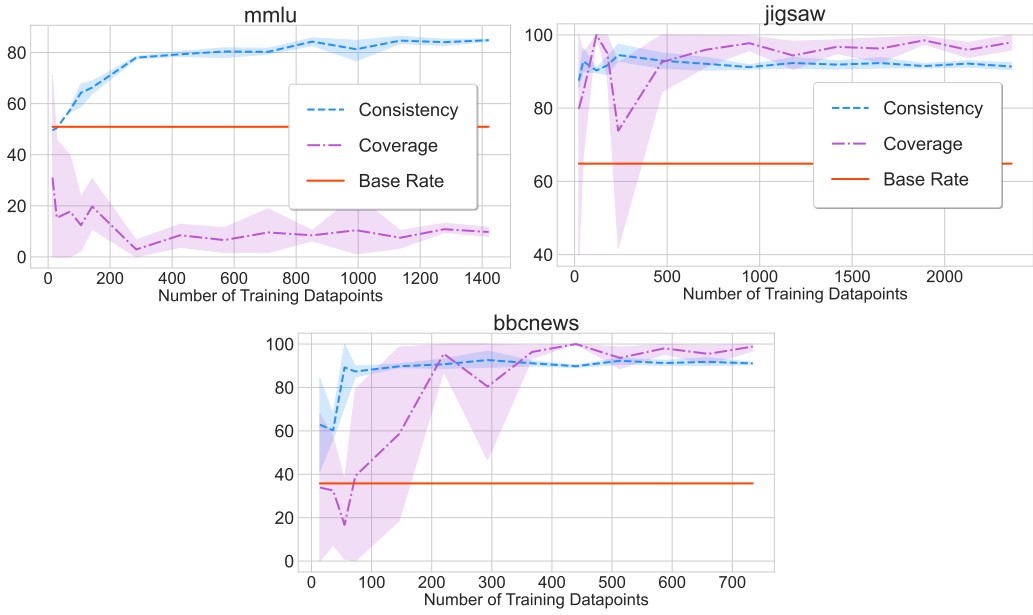

Figure 9: Within 500 training datapoints, probes often approach the consistency and coverage ($\alpha = 0.9$) they will attain when trained on the entire dataset.

### B.3 Confidence Threshold ablation

As a general trend (Figure 11), increasing the confidence threshold $\alpha$ has no impact until a point, after which it leads to a steady decline in coverage and a steady increase in estimation consistency.

## C Experiment Details

### C.1 Datasets:

We used the following datasets in our experiments, all usage is in accordance with their respective licenses. For each dataset, we select a maximum of 50,000 training instances to train (and validate) our probes, using the full test set to measure all metrics.

#### C.1.1 Multiple Choice Question Answering:

To test this, we collect 8 MCQ datasets—MMLU [34], CosmoQA [39], PiQA [14], ARC [21], MedMCQA [69], CommonsenseQA [85], OpenbookQA [63] and QASC [50] and use CoT to generate outputs with explanations before the answer (for more prompts see Appendix C).

**ARC:** The AI2 Reasoning Challenge (ARC) [21] is a knowledge and reasoning challenge that contains 7,787 natural, grade-school science questions (authored for human tests).

**CommonsenseQA:** The CommonsenseQA [85] dataset contains 12,247 questions with complex semantics that often require prior knowledge.

**MedMCQA:** A large-scale, Multiple-Choice Question Answering (MCQA) dataset designed to address realworld medical entrance exam questions, MedMCQA [69] has 194k AIIMS and NEET PG entrance exam MCQs covering 2.4k healthcare topics and 21 medical subjects.

**MMLU:** The Massive Multitask Language Understanding Benchmark [34] is a test to measure a text model's multitask accuracy. The test covers 57 tasks including elementary mathematics, US history, computer science, law, and more.

**OpenBookQA:** Modeled after an open book examination, OpenBOOkQA [63], OpenBookQA consists of 6000 questions that probe an understanding of elementary level science facts and their application to novel situations.

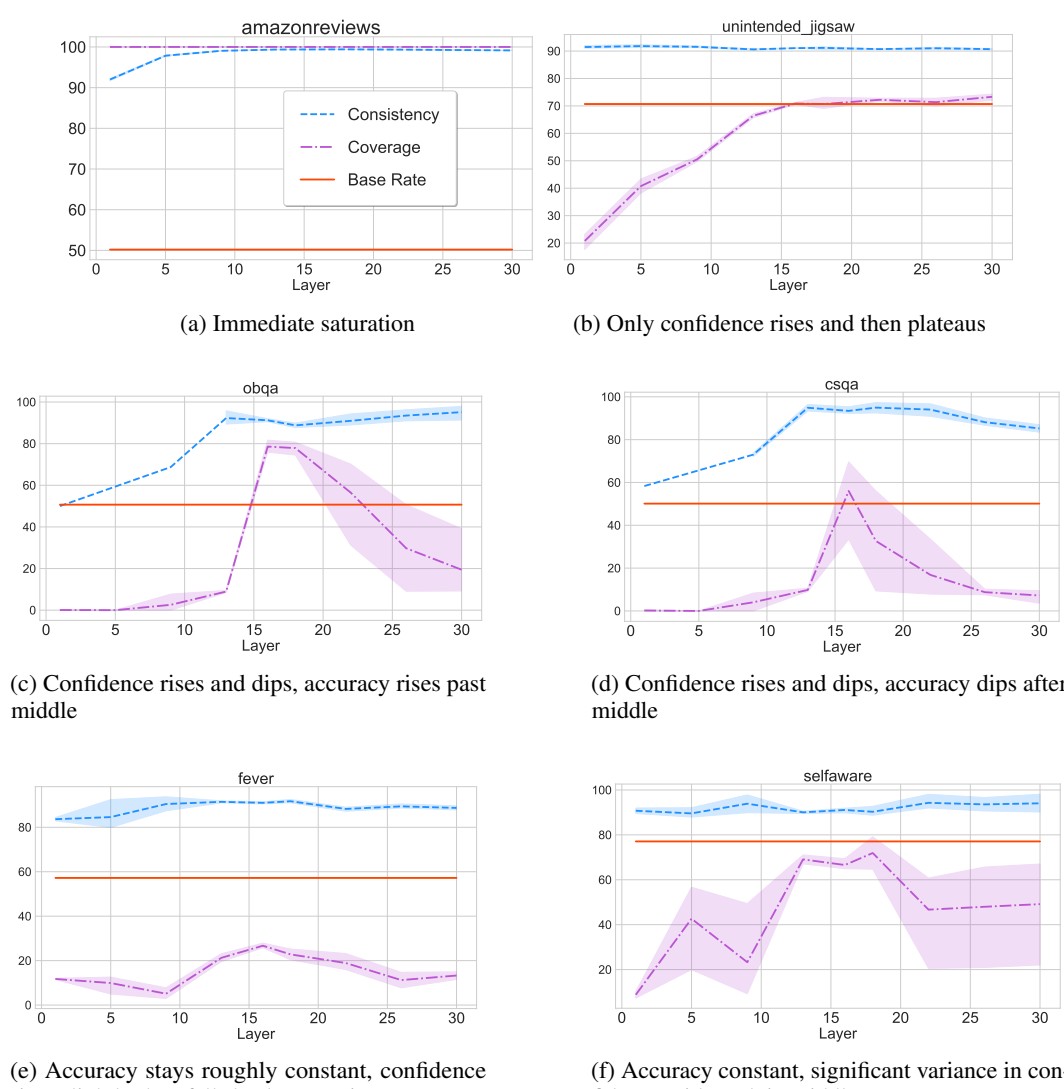

(a) Immediate saturation

(b) Only confidence rises and then plateaus

(c) Confidence rises and dips, accuracy rises past middle

(d) Confidence rises and dips, accuracy dips after middle

(e) Accuracy stays roughly constant, confidence rises slightly then falls back to starting

(f) Accuracy constant, significant variance in confidence with peak in middle

Figure 10: Layer ablations show that the decision on which layer to use is highly dependant on the specific task and dataset.

**QASC:** Question Answering via Sentence Composition [50] tests a models ability to compose knowledge from multiple pieces of texts. The dataset consists of 9,980 multiple-choice questions from elementary and middle school level science, with a focus on fact composition;

**PiQA:** Physical Interaction: Question Answering [14] is a dataset that tests whether AI systems can learn to reliably answer physical common-sense questions without experiencing the physical world. The dataset consists of 16,000 training QA pairs with 2,000 and 3,000 examples held out for validation and training.

**CosmoQA:** Commonsense Machine Comprehension Question Answering [39] is a dataset of 35,600 problems that require commonsense-based reading comprehension, formulated as multiple-choice questions. The dataset focuses on reading between the lines with questions that require reasoning beyond the exact text spans in the context.

### C.1.2 Sentiment Analysis:

**MTEB:** The following datasets were taken from the Massive Text Embedding Benchmark [64]. **AmazonReviews:** a dataset with 1.7 million Amazon product reviews and a sentiment score ranging

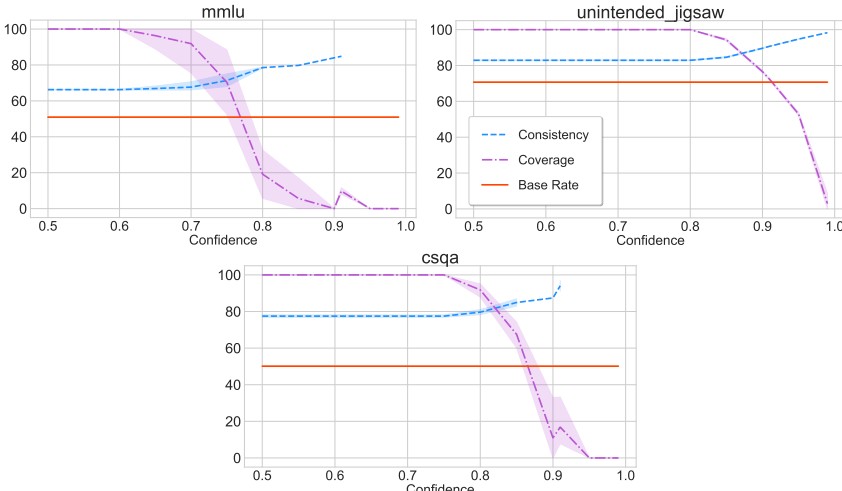

Figure 11: Increasing the confidence threshold $\alpha$ has no effect until a point, after which conformal consistency increases while the coverage tends to zero.

from 0-5, **TwitterSentiment:** a dataset with 30,000 tweets and a sentiment class from positive, neutral or negative

**Yelp Polarity:** The Yelp review rating challenge [7] consists of nearly 600,000 yelp reviews with sentiment classes from positive or negative.

**TwitterFinance:** The Twitterfinance dataset [2] is an annotated corpus of 11,000 finance-related tweets. This dataset is used to classify whether the tweet are bullish, bearish, or neutral.

**NewsMTC:** A dataset for sentiment analysis (TSC) on news articles reporting on policy issues, NewsMTC [33] consists of more than 11,000 labeled sentences.

**IMDB Reviews:** A classic sentiment analysis dataset, IMDB Reviews [60] consists of 50,000 highly polar movie reviews with binary labels.

**Financial Phrasebank:** A dataset that measures the polar sentiment [61] of sentences from financial news. The dataset consists of 4840 sentences from English language financial news categorised by sentiment. The dataset is divided by agreement rate of 5-8 annotators.

**AuditorSentiment:** Based on Financial Phrasebank, this dataset [40] is additionally annotated by auditors to reflect bearish, bullish and neutral labels for accounting related sentences.

**Emotion:** The DAIR-AI Emotion dataset [78] is a dataset of 20,000 Twitter messages with six basic emotions: anger, fear, joy, love, sadness, and surprise.

**SST5:** A standard sentiment analysis dataset, SST5 [82] consists of fine grained sentiment labels for 215,154 phrases in the parse trees of 11,855 sentences.

### C.1.3 Fact verification:

**ClimateFEVER:** A dataset that consists of 1,535 real-world claims regarding climate-change collected on the internet. Each claim In ClimateFEVER [23]is accompanied by five manually annotated evidence sentences retrieved from the English Wikipedia that support, refute or do not give enough information to validate the claim totalling in 7,675 claim-evidence pairs.

**HealthVER:** A dataset for evidence-based fact-checking of health-related claims, HealthVER [79] consists of 14,330 evidence-claim pairs.

**FEVER:** Fact Extraction and VERification [86] consists of 185,445 claims generated by altering sentences extracted from Wikipedia and subsequently verified without knowledge of the sentence they were derived from.

### C.1.4 Topic Identification:

**AGNews:** A collection of more than 1 million news articles. AGNews articles have been gathered from more than 2000 news sources and annotated for Topic [31].

**BBCNews:** The BBCNews dataset [30] is a dataset consisting of 2,225 articles published on the BBC News website corresponding during 2004-2005. Each article is labeled under one of 5 categories: business, entertainment, politics, sport or tech.

**NYTimes:** The New York Times Annotated Corpus [77] contains over 1.8 million articles written and published by the New York Times between January 1, 1987 and June 19, 2007 with article metadata provided by the New York Times Newsroom, the New York Times Indexing Service and the online production staff at nytimes.com.

### C.1.5 Toxicity Detection:

We use two datasets provided by the same organization, JigsawToxicity and JigsawUnintendedBias-Toxicity [1]. Both datasets have scores for toxicity from a chatroom setting.

### C.1.6 Others:

**Unanswerable Questions:** We use two datasets which contain unanswerable questions, Self-Aware [96] and KnownUnkown [5]. Selfaware is a dataset consisting of unanswerable questions from five diverse categories and their answerable counterparts. KnownUnknown is a dataset with questions on known quantities and unknown quantities. We collect jailbreaking prompts (and benign prompts) from WildJailbreak [45], a dataset with 262,000 vanilla (direct harmful requests) and adversarial (complex adversarial jailbreaks) prompt-response pairs. The dataset also has benign prompts that should be complied with.

**Format Following, Confidence Elicitation:** Both of these tasks use the same set of datasets - NaturalQA [54], TriviaQA [47] and MSMarco [65]. These datasets were selected as they are large question-answering datasets which often require generative output that can vary in length.

### C.2 Task Specific Set Up:

### C.2.1 Text Classification Tasks:

In all text-classification tasks, the CoT model is first asked to output a reasoning chain and then finally provide a classification answer. The probes are always trained to preemptively identify what the prediction will be using only the input token embeddings.

**MCQ:** For the MCQ tasks, we select two options by randomly sampling an incorrect answer from the provided options along with the correct option. This is because the MCQ datasets vary in terms of number of MCQ options, and for the out-of-distribution experiment we require all the datasets to have the same number of answer classes (note that the topic identification task below is multiclass, showing that our method is not limited to binary classification). The CoT model is asked to identify the correct answer option.

**Sentiment:** For the MCQ task, many datasets are in a binarized format, while others include continuous scores, or ternary labels (including neutral sentiment. We make all the datasets have a similar label structure, by keeping only positive and negative labels. Concretely, the label is 1 if and only if the sentiment is positive (or bullish for finance datasets), and 0 otherwise. The CoT model is asked to identify whether or not the text has a positive sentiment.

**Topic:** We keep only a subset of the topics to ensure every topic is fairly represented in the data. After dropping topics, AGNews has World, Sports, Businsess or Science, BBCNews has Business, Tech, Sports or Politics, while NYTimes has Health, Fashion, Real Estate or Television. The CoT model is asked to identify the topic of the article.

**Fact Verification:** We map all claims to with supported or not supported (includes refuted and neutral). For ClimateFEVER and HealthVer, we provide evidence along with the claim, while for FEVER we provide only the claim. The CoT model is asked to identify whether or not the claim is supported.

**Toxicity Detection:** We randomly sample the datasets to ensure a balance of toxic and not toxic comments. The CoT model is asked to detect whether or not the text is toxic

### C.2.2 Other Tasks:

In these tasks, the probes are trained to predict a variety of other behaviors using the input token embeddings.

**LM Abstention:** For the unanswerable question datasets of SelfAware and KnownUnknown, the model is first instructed to give a reasoning chain about the question, and answer only if the question is answerable (abstaining otherwise). With WildJailbreak, the input need not be a question, but any request. Hence, we instruct the model to comply with the request if it is not malicious, and abstain otherwise. We keep only the instances where the LM responds or complies, and train probes to detect whether or not the LM should have abstained (i.e. unanswerable question or malicious request).

**Format Following:** Using an input prompt, we specify two different output formats a LM must obey. In both cases, the probe learns whether the LM will fail to follow format specifications or comply with them.

- Bullets: The output should be presented in 3 numbered bullet points, no fewer and no more
- JSON: The output should be presented in a JSON string with the following structure: {['short_answer']: <str>, 'entities': List<str>, 'references': List<str>}

**Confidence Estimation:** There are two settings for this task:

- Internal Confidence: The LM is prompted to output an answer to the input question, and we record the token-normalized perplexity as a proxy of confidence. We keep only the bottom and top 25% of instances as per normalized perplexity and train the probes to differentiate between the two.
- Verbal Confidence: The LM is prompted to output an answer, along with a confidence score (either confident or unsure). The probe is trained to identify what the confidence score will be.

### C.3 Prompts and Inference:

We have written separate prompts for each dataset to ensure the LM follows the instructions and outputs the text in a way that guarantees we can parse it and infer the behavior we seek to pre-emptively identify with the probes. For example, a prompt for the MMLU dataset for MCQA is :

```
Question: What is true for a type-Ia supernova?
Option A: This type occurs in young galaxies
Option B: This type occurs in binary systems
Give an explanation and then the answer:
Explanation: Type Ia supernova is a type of supernova that occurs when
two stars orbit one another in which one of the stars is a white dwarf
Answer: B
Question: <NEW QUESTION>
Give an explanation and then the answer:
Explanation:
```

For the JSON task, one such prompt is:

```
Answer the following questions by giving a short_answer, entities list
 and references list. Give the output in JSON format
 Question: What is the capital of France?
 Answer: { "short_answer": "Paris",
 "entities": ["France"], "references": ["https://en.wikipedia.org/wiki/Paris"]}
 Question: <NEW QUESTION>
 Answer:
```

Each task and dataset has a different set of prompts. We have provided a link to the code, and all prompts can be seen in the file data.py

All LM inference uses greedy decoding and is hence deterministic.

### C.4 Hardware:

All of our experiments were run on a compute cluster with 8 NVIDIA A40 GPUs (approx 46068 MiB of memory) on CUDA version 12.6. The CPU on the cluster is an AMD EPYC 7502 32-Core Processor. Most experiments could be conducted with less than 16GB of GPU RAM.

Generating the hidden states took around 2 hours for every 10,000 points, while training a hidden state probe takes fewer than 30 seconds.

### C.5 Replicability:

To ensure that our code is easy to replicate and our method is easy to extend, we have provided open access to our code.

# NeurIPS Paper Checklist

The checklist is designed to encourage best practices for responsible machine learning research, addressing issues of reproducibility, transparency, research ethics, and societal impact. Do not remove the checklist: **The papers not including the checklist will be desk rejected.** The checklist should follow the references and follow the (optional) supplemental material. The checklist does NOT count towards the page limit.

Please read the checklist guidelines carefully for information on how to answer these questions. For each question in the checklist:

- You should answer [Yes] , [No] , or [NA] .
- [NA] means either that the question is Not Applicable for that particular paper or the relevant information is Not Available.
- Please provide a short (1–2 sentence) justification right after your answer (even for NA).

**The checklist answers are an integral part of your paper submission.** They are visible to the reviewers, area chairs, senior area chairs, and ethics reviewers. You will be asked to also include it (after eventual revisions) with the final version of your paper, and its final version will be published with the paper.

The reviewers of your paper will be asked to use the checklist as one of the factors in their evaluation. While "[Yes] " is generally preferable to "[No] ", it is perfectly acceptable to answer "[No] " provided a proper justification is given (e.g., "error bars are not reported because it would be too computationally expensive" or "we were unable to find the license for the dataset we used"). In general, answering "[No] " or "[NA] " is not grounds for rejection. While the questions are phrased in a binary way, we acknowledge that the true answer is often more nuanced, so please just use your best judgment and write a justification to elaborate. All supporting evidence can appear either in the main paper or the supplemental material, provided in appendix. If you answer [Yes] to a question, in the justification please point to the section(s) where related material for the question can be found.

IMPORTANT, please:

- **Delete this instruction block, but keep the section heading "NeurIPS Paper Checklist",**
- **Keep the checklist subsection headings, questions/answers and guidelines below.**
- **Do not modify the questions and only use the provided macros for your answers**.

