# OpenReview forum: "Language Models Can Predict Their Own Behavior"
_NeurIPS.cc/2025/Conference — NeurIPS 2025 poster_

### Official Review · Reviewer_fHv2 · 2025-06-07

**Clarity:** 3
**Significance:** 4
**Originality:** 2
**Rating:** 4
**Confidence:** 4

**Summary:**

This paper shows that large language models (LLMs) often encode signals in their internal states that can predict their future behavior before generating any output. By training simple linear probes on the hidden representations of input tokens and calibrating them with conformal prediction, the authors build early warning systems for issues like instruction-following failures, safety alignment violations (e.g., jailbreaking), and low-confidence responses. These probes can reduce harmful outputs and improve efficiency, such as cutting inference costs by 65% in Chain-of-Thought prompting with minimal accuracy loss. The approach is lightweight, data-efficient, and scales well with larger models. It generalizes to unseen tasks and helps reveal what LLMs implicitly "know" about their own future actions.

**Questions:**

1. In Figure 2, the performance gain of probing over BERT appears marginal. While conformal prediction improves probe accuracy, it’s unclear whether similar techniques could also enhance BERT performance.

2. In Section 4.1, the authors state they filter the training set to only include cases where the LM answers or complies. This risks making responses too similar, effectively turning the task into prompt classification—an issue discussed in the Weakness section. A better setup may involve using similar prompts that lead to different behaviors, training probes to predict those outcomes.

3. In Section 5, it would be useful to report the LLM’s accuracy without CoT. If it's comparable to CoT accuracy, the benefit of using probes to skip CoT may be limited.

**Ethical Concerns:**

["NO or VERY MINOR ethics concerns only"]

**Final Justification:**

The rebuttal’s empirical results are persuasive and address my main concern. But I agree with Reviewer DPE7 that the paper could benefit from narrowing its scope slightly to better fit within the 9-page limit.

**Limitations:**

yes

**Quality:**

3

**Strengths And Weaknesses:**

## Strengths:
1. The paper is well-written and easy to read.
2. The research topic is compelling—detecting when an LLM is "lying" or misbehaving by analyzing its internal states before token generation.
3. The paper effectively operationalizes the concept of "LLM predicting its own behavior" into practical techniques for improving safety and inference efficiency.

## Weakness:
1. My main concern is that high probing accuracy doesn't necessarily mean the LLM is predicting its own behavior. It could instead be classifying the input prompts, with the latent representations capturing prompt features rather than future behavior. While the BERT baseline partially addresses this, the performance gap between probing and BERT could also stem from architectural differences or parameter counts. **Although BERT is larger than the probing layer, the authors should take into account the parameter counts of the frozon LLM layers generating the latent representations. A more reasonable baseline would be using one causal LM of similar parameter counts to predict the behavior of another.** If such cross-prediction fails completely, I would be more convinced that the LLM is genuinely predicting its own behavior before token generation, and linear probing does capture the exact representations for prediction. I leave more questions about this in the Question section.

2. It would be helpful for the authors to discuss related concurrent work. For example, [1] also uses linear probing to predict LLM response features (e.g., token count, reasoning steps) before generation begins.

[1] Dong, Zhichen, et al. "Emergent Response Planning in LLM." arXiv preprint arXiv:2502.06258 (2025).

---

> ### Author Rebuttal · Authors · 2025-07-28
>
> We think your primary concern is an important one, and we would like to point out some ways that we directly address the question: "Are probes simply identifying features of the input, or are they truly predicting the future behavior of the LM?"
>
> **Several tasks in our study are impossible to perform by simply identifying features of the input:**
>
> We point to the tasks of format following (specifically the 3 bullet points task, which predicts whether a LM will output an answer in exactly 3 bullet points) and the jailbreak avoidance task (which predicts not whether an input text is malicious, but whether or not a language model will fail to abstain on a given malicious request). On the jailbreak avoidance datasets, LLama3-8B, Mistral-7B, and DeepSeek-R1 all have different abstention rates:(75%, 78% and % on the SelfAware dataset) and, in general, do not abstain/fail to abstain on the same input instances. We also see that probes trained on one LM **do not transfer to unseen LMs** (on the same dataset), i.e. a conformal probe trained and calibrated to provide 90% accuracy on the SelfAware dataset with LLama3-8B achieves only 60% accuracy on the Mistral-7B model. This shows that the probes are not identifying general, fixed features of the input text, but rather identifying model-specific patterns.
>
> **Pretrained LMs are incapable of predicting each others behavior in this way (without tuning):**
>
> We stick to the jailbreak avoidance task and perform the following experiment:
>
> Llama3-8B is used to generate hidden states, and we then annotate the instance for whether or not Llama3 failed to abstain from a malicious input. We then measure the performance of the following methods:
> Probe trained on the hidden states to predict whether or not LLama3 will fail to abstain
> FewShot Prompting of Mistral7B to predict whether or not LLama3 will fail to abstain i.e. without any retraining / finetuning
>
> The performance is as shown below:
> | Dataset      | Random | Probe | Mistral (No FT) |
> |--------------|--------|-------|-----------------|
> | KnownUnknown | 80.31  | 89.46 | 75.81           |
> | SelfAware    | 84.4   | 97.66 | 78.91           |
>
> As we can see, when we do not fine-tune Mistral for the task, it performs worse than random, and significantly worse than the probes. This suggests that the property we are trying to predict is not one that can be inferred from just the text alone and requires some model-specific information to reliably ascertain.
>
> **Directly addressing the suggested experiment:**
>
> We finally address the experiment suggested by the reviewer: "A more reasonable baseline would be using one causal LM of similar parameter counts to predict the behavior of another. If such cross-prediction fails completely, I would be more convinced that the LLM is genuinely predicting its own behavior before token generation, and linear probing does capture the exact representations for prediction."
>
> We have shown above that such cross-prediction fails when we are not allowed to train the other causal LM (Mistral); however, we suspect the reviewer was more curious about the case where you train the other causal LM to predict this. We would maintain that in such an experiment, you **should not** expect the cross-prediction to fail.
>
> This is because, when provided with sufficient training data in the form of (input text, output behaviour), the other causal language model will eventually learn the patterns present in the data, and hence learn the behaviour of the LLama model. This would not be evidence that the probes are simply identifying input features, but rather evidence that with sufficient training data and a sufficiently powerful text classification system (in this case, Mistral with a classification head), you will be able to model the patterns in the data and learn the Language Model's behavior.
>
> The point of our paper is to show that you do not need to distil the Llama model in this way to get a reliable signal on its behavior, because the hidden states provide you with helpful information.
>
> However, we understand the core concern here, and so we perform a modified experiment to demonstrate our point. We compare the probes to Lora-finetuning Mistral in a **data scarce** scenario: i.e. we train probes and the Mistral model to identify whether or not LLama3 will fail to abstain using only 50 datapoints. The performance is as follows:
>
> | Dataset       | Random | Probe | Mistral |
> |---------------|--------|-------|---------|
> | KnownUnknown  | 80.31  | 84.6  | 81.27   |
> | Selfaware     | 84.4   | 89.0  | 80.53   |
>
> While the Mistral finetune is better than few-shot prompting, showing that some pattern is being established, it is worse than the probe. This data-efficient learning ability from the probe shows that the pattern is more readily readable from the hidden states, which shows the core claim of our paper: that hidden states of LMs contain information on how the LM will behave.
>
>
> **On weakness: discuss concurrent work**
>
> We thank the author for bringing this related work to our attention. We have added writing for both this and the work that it cites into the related work section of our paper.

---

> > ### Comment · Reviewer_fHv2 · 2025-08-03
> >
> > The rebuttal’s empirical results are persuasive and address my main concern. But I agree with Reviewer DPE7 that the paper could benefit from narrowing its scope slightly to better fit within the 9-page limit. That said, I’m happy to raise my score to 4.

---

> > > ### Author Response · Authors · 2025-08-04
> > >
> > > Thank you for your response and for agreeing to raise the score. Please do update the review to reflect this. If the paper is accepted, we will ensure that we use the contents of this discussion to clarify the presentation of our ideas, keeping your initial concern in mind.

---

### Official Review · Reviewer_74n8 · 2025-06-24

**Clarity:** 3
**Significance:** 2
**Originality:** 3
**Rating:** 4
**Confidence:** 4

**Summary:**

This paper introduces a novel and practical method for predicting the future behavior of a language model by probing the internal representations of its input tokens, before any text is generated. By integrating simple linear probes with conformal prediction, the authors create a reliable early-warning system with provable error bounds. The method is validated across a wide range of applications, including preemptively detecting safety failures (e.g., jailbreaking) and accelerating Chain-of-Thought inference, demonstrating significant performance gains (e.g., 91% jailbreak reduction, 65% inference cost reduction) with minimal overhead and strong generalization.

**Questions:**

- The finding that "bigger models lead to better probe performance" is a fascinating one. Is it because larger models encode their future behavioral plans in a more "linearly separable" or "redundant" manner? Does this performance improvement trend continue indefinitely, or does it plateau?

**Ethical Concerns:**

["NO or VERY MINOR ethics concerns only"]

**Final Justification:**

The author's rebuttal has mainly addressed my technical concerns, but the resposne to my question of reason for better probe performance in bigger models is missing. So I will maintain my score.

**Limitations:**

yes

**Quality:**

3

**Strengths And Weaknesses:**

### Strenghts
- The core idea of preemptive behavior prediction is innovative and provides insights into the critical, real-world problems of high inference costs and the need for proactive safety guardrails.
- The claims are substantiated by extensive experiments across a diverse set of 27 datasets with OOD generalization tests.

### Weaknesses
- The paper primarily probes the hidden state of the final input token. The motivation and how does the attention mechanism effectively compress the "intent" or "future behavioral plan" of the entire prompt into this single token's state are unclear.
- Apart from the out-of-distribution (OOD) generalization tested across different datasets of the same task type, it is necessary to discuss this generalization against more fundamental distribution shifts (e.g., code vs. text classification).
- The authors chose simple linear probes, which ensures interpretability. However, this might limit their ability to capture more complex, non-linear behavioral patterns. Could a small, non-linear probe (e.g., a shallow MLP) unlock predictions for more nuanced behaviors (e.g., sarcasm, humor) without excessively increasing computational overhead?
- Lack analysis of the overhead introduced by collecting hidden states for probe training.

---

> ### Author Rebuttal · Authors · 2025-07-27
>
> **On weakness: Probing final input token with unclear motivation**
>
> * Our decision to use the final token is motivated by a large body of work[1, 2, 3], which shows that the final token encodes useful information that is relevant for the prediction of a variety of properties of the input and output. Hence, we consider our choice of using the final input token as well motivated.
> * We also provide an analysis on the performance of using tokens at different positions, and show that the final token does contain more useful representations for behavior prediction, further solidifying this growing observation in the field
> * We understand that there are potentially methods of mechanistic interpretability that could be deployed to understand why the final token encodes the most information, however that is beyond the scope of this paper. Given the page limit, we would ask the reviewer to consider that the other sections we have added are of a more direct relevance to the paper, and we leave a deep understanding of the mechanism of attention, compressing the intent to future work.
>
>
> **On weakness: Lack of analysis of the overhead introduced by collecting hidden states for probe training**
>
> 1. We would first point out that any method which hopes to learn the patterns of the LMs behaviors require some source of training data, and the cost of generating that data is roughly equal. For example, in order to generate the probe training data, we must run our LM on a training set, collect the hidden states and collect the output. In order to train our BERT baseline, we must collect the input text and the output, a process which naturally generates the hidden states as a result of the forward pass. Hence, the only extra cost of the probe (when compared to the BERT baseline), is that the hidden states must be stored in memory. This is not a significant cost, and is more than offset by the cost of fine-tuning BERT. Additionally, considering that once the probe has been trained, the hidden states can then be deleted, we do not consider this to be a significant overhead.
> 2. Furthermore, we show (Figure 6) that within 500 training datapoints, probes often approach the consistency and coverage they will attain when trained on the entire dataset. This suggests that the amount of training data required for good probe performance is small, minimizing the cost of training.
>
>
> **On weakness: The authors chose simple linear probes...**
>
> This is an interesting question. We experiment with a variety of probe models, comparing Differences in Means, Linear Models, shallow MLPs, and RandomForests.
>
> | Dataset      | Random | Difference in Means | Linear | RandomForest | MLP    |
> |--------------|--------|---------------------|--------|--------------|--------|
> | MMLU         | 51.36  |  35.39              | 69.51  |  61.81       |  71.21 |
> | KnownUnknown | 74.95  |  26.22              |  81.92 |  82.43       |  83.81 |
> | SelfAware    | 78.92  | 19.08               |  83.85 |  83.54       |  87.08 |
>
> The Linear and RandomForest approaches have comparable performance, while a shallow MLP proves consistently more capable. We believe that on KnownUnknown and SelfAware, the difference is not drastic enough to enable nuanced behavior detection, however, on MMLU, the improvement is significant. As you mention, even more sophisticated probes have the potential to identify more nuanced behavior, but such a decision would increase computational overhead significantly. In our work, we focus on linear probes primarily for their simplicity, however we do agree that there are several options that strike different points on the compute-performance curve, and there are multiple options in the low-compute regime that can give good performance.
>
> **On weakness: It is necessary to discuss generalization against more fundamental distribution shifts (e.g., code vs. text classification).**
>
> We would first remember the overall goal of the probe: to preemptively predict a specific behaviour of the LM. When the distribution is shifted so fundamentally that the behaviours you are trying to predict is not well defined, it does not make sense to even try to apply a probe trained on one distribution on the other. For example, if you have trained a probe to preemptively predict which text classification option a LM will choose, there is no sense in applying it to a LM that is trying to synthesise a Python program (as the behaviour of interest, selecting a specific option, does not occur in this distribution). We do not see this as a limitation of our method, as no method that seeks to predict a behaviour can generalise to distributions where that behaviour itself is not well defined. What we have shown is that when this behaviour is well defined (ensured by taking multiple datasets spanning different domains, but having some common task properties), then we see signs of OOD generalisation.
>
> We understand the importance of showing that our probes can generalize to fundamentally unseen domains, and conduct a shifting domain experiment to verify this. We stick within the sentiment prediction task, and use only news-based, generic sentiment datasets to train the probes (twitter-mteb and news-mtc). Then, we test our probes on financial sentiment analysis datasets, a shifted domain. The results show that despite never explicitly training on such financial data, the probes are able to achieve significantly higher than random performance:
> | Dataset              | Random | Probe  |
> |----------------------|--------|--------|
> | Twitter-Finance      | 51.8   |  77.69 |
> | Financial Phrasebank | 54.52  | 96.32  |
>
>
> [1]: Li, Kenneth, et al. "Inference-time intervention: Eliciting truthful answers from a language model." Advances in Neural Information Processing Systems 36 (2023): 41451-41530.
>
> [2]:  Azaria, Amos, and Tom Mitchell. "The internal state of an LLM knows when it's lying." arXiv preprint arXiv:2304.13734 (2023).
>
> [3]: Von Rütte, Dimitri, et al. "A language Model's guide through latent space." arXiv preprint arXiv:2402.14433 (2024).

---

> > ### Comment · Reviewer_74n8 · 2025-08-06
> >
> > Thanks for your rebuttal. My concerns are mostly addressed. I will keep my score.

---

### Official Review · Reviewer_hGPd · 2025-07-03

**Clarity:** 3
**Significance:** 2
**Originality:** 2
**Rating:** 4
**Confidence:** 2

**Summary:**

This paper proposes conformal probes, simple linear classifiers trained on a language model’s input-token hidden states and calibrated with conformal prediction. Once calibrated, the probes fire only when they can guarantee a user-specified error rate, giving an early-warning or early-exit signal before the model generates a single output token.
- Safety & alignment: Detect prompts that will cause refusal-policy failures, cutting jailbreak success from 30 % to 2.7 % (≈ 91 % relative reduction) on the WildJailbreak set.
- Format & instruction following: Predict whether an answer will violate output-format constraints with higher precision than a 340 M-parameter fine-tuned BERT baseline while using only ~0.0025 % as many trainable parameters.
- Efficiency: Interrupt Chain-of-Thought (CoT) generation for text-classification tasks when the probe is confident, yielding a 65 % average reduction in forward passes across 27 datasets with ≤ 1.4 % worst-case accuracy loss and often a small accuracy gain

**Questions:**

1. Scope of detectable behaviours
- You report that probes cannot predict outright answer correctness in MCQ tasks. Could you quantify this ceiling (e.g., accuracy vs. random) and elaborate on what differentiates behaviours the probe can vs. cannot capture? Clarifying this boundary would strengthen Section 6.

2. Latency & resource overhead
- While the method reduces generation compute, each query still requires a forward pass to produce hidden states for the probe. Please provide wall-clock latency and energy measurements relative to vanilla inference to validate net savings.

3. Calibration stability across domains
- How sensitive are the conformal thresholds to domain shift? The OOD CoT results are encouraging, but would thresholds learned on, say, conversational data hold for code or medical text? A small experiment or guidance on recalibration frequency would help practitioners.

4. What are the practical advantages and potential drawbacks of turning conformal probes into a proprietary, in-house safety and efficiency layer?

5. How would conformal probes perform differently on reasoning‐focused LMs versus instruction‐tuned LMs, and what model-specific considerations should be taken into account?

**Ethical Concerns:**

["NO or VERY MINOR ethics concerns only"]

**Final Justification:**

latency data, broader model tests, and the conformal guarantee clarification address my concerns, so I’m raising my score to 4. It’s clear that behavior-level early exits offer a valuable safety-efficiency layer.

**Limitations:**

yes.

**Paper Formatting Concerns:**

None.

**Quality:**

3

**Strengths And Weaknesses:**

## Strengths
- Writing is concise.
- A lightweight, model-agnostic safety-efficiency layer that can sit atop any inferene stack and composes with speculative decoding; addresses growing cost of long generations.

## Weakness
- Experiments are somewhat limited to the Llama3 families.
- No latency measures for probe inference itself.
- Linear probing are conceptually simple; early-exit ideas exist in speculative decoding (or adative inference) over vision/nlp. So, somewhat incremental novelty may hinge on the conformal guarantee.

---

> ### Author Rebuttal · Authors · 2025-07-28
>
> **On question: Practical advantages / potential drawbacks**
>
> We think that conformal probes can be a significant boon to organisations hoping to limit the computational cost of their LM inference. By identifying easy instances, catching malicious inputs early and exiting when they have precisely anticipated the LMs output, the probes can greatly reduce time and cost, while giving users exact control over the allowable error rate.
>
> Conformal probes are not without their limitations. Specifically, the mathematical guarantee on the error rate only holds when the test data is exchangeable (similar to IID), and hence if deployed on data that is too different from the training distribution, it may fall below the user's stated performance threshold without notice. We speak more about this below, in an answer to your question on conformal calibration, where we suggest ways to use this method with confidence even in the OOD setting.
>
> **On question: latency and overhead of inference efficiency measurement**
>
> We thank you for your attention to this detail. We would note our initial motivation for measure % of tokens generated as opposed to wallclock time: we wanted a measure of efficiency that is agnostic to specific hardware implementations, and that captures the reason why the efficiency of the probe-based generation is superior. However, we understand your concerns that the empirical reality may not match up with this more conceptual metric, hence we measure the wallclock time of normal inference v.s. Inference with conformal probes guided early exiting on Llama3-8B using alpha=0.8, and present the results in the table below:
>
> | Dataset      | Total Inference Time (normal) | Total Inference Time (conformal probes + early exit) | Inference time reduction (%) | % Early Exit | Token generated reduction (%) |
> |--------------|-------------------------------|------------------------------------------------------|------------------------------|--------------|-------------------------------|
> | Selfaware    | 2600                          | 169                                                  | 93.5                         | 100          | 96.8                          |
> | KnownUnknown | 4131                          | 380.9                                                | 90.77                        | 99.84        | 92.4                          |
> | MMLU         | 1,12,336                      | 93,598.5                                             | 16.67                        | 17.24        | 15.91                         |
>
>
> As we can see, the inference cost reductions are significant even when measured by wallclock time. Importantly, the inference time reduction roughly matches the % reduction in tokens generated, which is the original metric we report. This confirms that empirical results match our more hardware-agnostic measures of inference efficiency.
>
> **On question: Calibration stability across domains**
>
> This is an important question, and we would like to answer it from two perspectives:
> 1. Theoretically, no, the learned conformal thresholds do not apply under shifting domains. This is intuitive as the formal guarantee that the threshold will bound the error rate depends on the test dataset being exchangeable (similar to IID) with the calibration set. This does not hold in the OOD case, and hence, unless we add assumptions on the statistical nature of the distribution shift (which is hard to do for abstract concept shifts like code v.s. medical text) we cannot guarantee a bound on the error rate.
> 2. Emperically, we see optimistic results. While there is significant variance across tasks and datasets, we observe that the calibration threshold still limits the error rate when used in the ood setting. To see this, refer to Figure, which uses alpha=0.9 and hence hopes to limit the error rate to 10%. We see that across 18 datasets, this threshold is held even in the OOD setting. However, from our experiments we are able to give the following recommendations for practitioners who hope to use conformal probes in an OOD setting:
> a. As usual, the training set that is closest in approximation to the testing set will give the most secure guarantees for the threshold. This is seen in the case of the sentiment datasets: we use only news-based, generic sentiment datasets to train the probes (twitter-mteb and news-mtc). Then, we test our probes on both generic (IMDB) and financial (TwitterFinance) sentiment analysis datasets, a shifted domain. The results show that when we use news based hold out datasets, the calibration is more precise
> | Dataset         | Conformal Consistency with alpha=0.8 | Conformal Consistency with alpha=0.9 |
> |-----------------|--------------------------------------|--------------------------------------|
> | Twitter-Finance |  77.69                               | 81.21                                |
> | IMDB            | 83.72                                | 86.81                                |
>
>
> b. When in doubt, having more datasets from a particular task is helpful. We see this in the MCQA task, where calibration is consistently better when we add more OOD datasets into the training set (for a single, fixed test dataset). This suggests that include a diverse range of training data helps improve calibration
> c. Expect larger than specified errors, and if the deployment scenario is safety critical, then use a small hold out test set to validate the conformal probes performance before using it in practice.
>
> We believe that the conformal probes show promising signs of generalizing to unseen distributions, and that a careful selection of training datasets can provide highly performant and well calibrated probes.
>
> **On question: scope of detectable behaviors**
>
> We are quite interested in better understanding the kinds of behaviors that LM hidden states struggle to predict. One such behavior is whether or not the LM will incorrectly answer an MCQA question. To quantify this, we experimented with 8 MCQ datasets, and trained hidden state probes to predict whether or not the LM will output an incorrect answer to a question. On 5/8 of these datasets, the hidden probes (when used without any conformal calibration) perform worse than, or equivalent to random performance. On the other 3 datasets, the probes perform slightly better than random, however any attempts at calibration show that no reliable threshold can guarantee an error rate of less than 60%, which is quite poor.
>
> More generally, we find that there are two types of tasks that LM hidden states cannot predict well:
> 1. Lexical quantification tasks: We experimented with tasks such as, predict the exact number of words that will be in the output, predict the number of verbs that will appear in the output etc. While the probes are able to perform reasonably (considerably higher than random) on the binarized version of these tasks (i.e. will the output be long or short, will the output contain a verb or not), it performs poorly on the quantification version. This is consistent with prior works that show LMs struggling with counting based problems[1], and suggests that the problem transfers to hidden state representations.
> 2. Tasks which require external knowledge: Our hypothesis as to why the MCQ tasks fail, is that unlike the other tasks we explore of format following, jailbreaking, etc, external knowledge is required to verify an answer to the MCQ questions. We can be sure that this knowledge is either not present, or hard to access in the LM, as it incorrectly answers many of these MCQ questions, which suggests that the hidden probes will always struggle to detect when the LM is going to incorrectly answer.
>
>
> **On Weakness: Experiments limited to LLama3**
>
> We have made sure to include ablations on other models in the Appendix, showing that the trends discussed in the main text are replicable on the Mistral-7B and DeepSeek-R1-Qwen-14B models. We follow this up with additional experiments below, we train hidden state probes on the following model families: Mistral (Mistral-7B), Qwen3-8B, DeepSeek-R1
>
> The probe results (when used without any conformal calibration) shows that we achieve above random performance with all LMs, showing the central claim of our work: that the hidden states of LMs can predict the future behavior of the output.
> | Dataset      | Random | Llama3-8B | Mistral-7B | DeepSeek-R1 | Qwen3-8B |
> |--------------|--------|-----------|------------|-------------|----------|
> | MMLU         | 51.36  | 69.51     |  69.9      | -68.45      | -71.42   |
> | KnownUnknown | 74.95  |  81.92    | 85.08      | 75.96       | 81.12    |
> | SelfAware    | 78.92  |  83.85    |  88.74     | 79.92       | 82.62    |
>
> We observe that when using conformal alpha = 0.90, the probes are able to maintain high estimation consistency regardless of the underlying LM used, showing that the method is able to properly calibrate probes for a variety of LMs.
>
> | Dataset      | Random | Llama3-8B | Mistral-7B | DeepSeek-R1 | Qwen3-8B |
> |--------------|--------|-----------|------------|-------------|----------|
> | MMLU         | 51.36  | 89.10     |  88.15     | 89.45       | 91.43    |
> | KnownUnknown | 74.95  | 93.33     | 88.64      | 92          | 89.55    |
> | SelfAware    | 78.92  | 91.21     | 89.06      | 83.5        | 89.94    |
>
> Our motivation for focusing on the Llama3 family in the main text is primarily to keep the focus on the methods and tasks, as opposed to the specific language models used. We would posit that we have shown sufficient evidence to believe that none of our results are tied to any particular feature of the Llama3 LMs, and would hence disagree that the paper's use of Llama3 is a significant weakness.
>
>
> [1]: Fu, Tairan, et al. "Why Do Large Language Models (LLMs) Struggle to Count Letters?." arXiv preprint arXiv:2412.18626 (2024).

---

> > ### Author Response · Authors · 2025-08-03
> > **Requesting response**
> >
> > We thank the reviewer for their efforts in reviewing our work and would like to ask whether they found our rebuttal satisfactory.
> >
> > There were 3 main weaknesses:
> > 1. Experiments are somewhat limited to the Llama3 families.-> Addressed above
> > 2. No latency measures for probe inference itself. -> Addressed above with wallclock measurements. Note that since the probe is a linear layer with no activation, its inference introduces very little latency. This is seen in the results above.
> > 3. Linear probing are conceptually simple; early-exit ideas exist in speculative decoding (or adative inference) over vision/nlp. So, somewhat incremental novelty may hinge on the conformal guarantee. -> While we agree that linear probing is simple, we disagree that our form of early-exit is covered in existing work of speculative decoding / adaptive inference. The majority of these methods exit early on the prediction of an output token, and exit early if they can estimate the specific tokens a LM will output. This is meaningfully different from predicting behavior, which often requires the entire output sequence to judge (see format following tasks), and cannot be judged from output tokens alone (see confidence task and jailbreaking task).
> >
> > Additionally, there were a few questions posed:
> > 1. Scope of detectable behaviours -> Addressed above
> >
> > 2. Latency & resource overhead -> Addressed above with wallclock measurements
> >
> > 3. Calibration stability across domains -> Addressed above with domain transfer experiment
> >
> > 4. What are the practical advantages and potential drawbacks of turning conformal probes into a proprietary, in-house safety and efficiency layer? --> Addressed above
> >
> > 5. How would conformal probes perform differently on reasoning‐focused LMs versus instruction‐tuned LMs, and what model-specific considerations should be taken into account? --> Addressed above with comparison between LLama3 and DeeepSeek-R1
> >
> > We believe we have answered the core concerns the reviewer has expressed, and ask that they respond. If our reply is unsatisfactory in some way, please let us know so we may have the time to follow up.

---

> > > ### Author Response · Authors · 2025-08-05
> > >
> > > We would like to remind the reviewer that there only three days of the discussion period remaining, and would like to request that they participate in the discussion.
> > >
> > > We have made a good faith attempt to address the majority of your concerns. If you find our responses lacking, please do let us know so that we may directly discuss them in time.

---

### Official Review · Reviewer_DPE7 · 2025-07-04

**Clarity:** 2
**Significance:** 3
**Originality:** 3
**Rating:** 4
**Confidence:** 4

**Summary:**

This paper explores the concept of enabling models to predict their own behavior in advance. The proposed method aims to identify potentially problematic behaviors a priori within a given confidence level, thereby facilitating timely intervention. The authors demonstrate the applicability of their approach across multiple tasks, including instruction following, safety alignment, uncertainty estimation, and inference acceleration. While the idea of a priori behavior prediction is promising, the paper would benefit from a stronger focus on a specific domain or core problem. The current broad coverage makes it challenging to evaluate the depth and significance of the technical contribution.

**Questions:**

It is unclear from the paper whether the backbone model is updated during the training of the classifiers. If the backbone is fine-tuned, it might lead to better probing performance, but could also affect the language model’s generation quality. What is the trade-off between probe effectiveness and model behavior stability in such a setting?

**Ethical Concerns:**

["NO or VERY MINOR ethics concerns only"]

**Limitations:**

Yes

**Paper Formatting Concerns:**

-

**Quality:**

2

**Strengths And Weaknesses:**

**Strengths**

-	The idea of a priori behavior prediction has strong practical and application value.
-	The paper is well-written and easy to understand.

**Weaknesses**

-	The paper feels disorganized and lacks focus. It covers a wide range of topics—instruction following, safety alignment, uncertainty estimation, and inference acceleration—without clearly defining its main objective. These topics should be explored in separate studies rather than being briefly touched upon in a single paper.
-	For most of the domains discussed, there is little technical distinction between attaching a classifier to a large model and using BERT-based classification; the experimental results also reflect this lack of differentiation.
-	The paper does not compare or discuss its approach against post-hoc decision-making methods, which limits the evaluation of the proposed method's advantage.

**Minor Issues and Suggestions**

-	Figure 2 may be misleading in its current presentation, even though the coverage percentages are explained in the caption. The term Estimation consistency is also inaccurately used — it should refer specifically to the estimation accuracy among the covered samples , not overall.
-	In the discussion of inference acceleration, the cost of inference is not solely determined by the number of generated tokens; the number of input tokens should also be taken into account. Therefore, the relevant claims or reported metrics should be revised accordingly to reflect a more accurate cost analysis.

---

> ### Author Rebuttal · Authors · 2025-07-28
>
> **On weakness: Paper covers a wide range of topics**
>
> We thank you for your feedback and would like to ask you for suggestions on how we may rewrite certain sections to make the paper feel more organised.
> However, we would expressly disagree that the wide range of topics covered is a weakness. This paper is not attempting to set the state-of-the-art in a particular task, but rather is trying to demonstrate a phenomenon that occurs in LMs across a wide range of inputs. That is, we are trying to firmly establish that across a wide range of inputs and behaviours, the hidden states of LMs contain valuable information on how the model will behave. We additionally hope to show that by leveraging this information, we can craft efficient and precise early warning and exit systems for these behaviours. The fact that the same method of conformal probes is applicable to such a diverse range of topics without any modification is one of its greatest strengths, and shows the generality of our findings. We do not agree that the work should have instead focused on a single set of tasks.
>
> **On issue: cost of inference measurement**
>
> We thank you for your attention to this detail. We would note our initial motivation for measure % of tokens generated as opposed to wallclock time: we wanted a measure of efficiency that is agnostic to specific hardware implementations, and that captures the reason why the efficiency of the probe-based generation is superior. However, we understand your concerns that the empirical reality may not match up with this more conceptual metric, hence we measure the wallclock time of normal inference v.s. Inference with conformal probes guided early exiting on Llama3-8B using alpha=0.8, and present the results in the table below:
>
> | Dataset      | Total Inference Time (normal) | Total Inference Time (conformal probes + early exit) | Inference time reduction (%) | % Early Exit | Token generated reduction (%) |
> |--------------|-------------------------------|------------------------------------------------------|------------------------------|--------------|-------------------------------|
> | Selfaware    | 2600                          | 169                                                  | 93.5                         | 100          | 96.8                          |
> | KnownUnknown | 4131                          | 380.9                                                | 90.77                        | 99.84        | 92.4                          |
> | MMLU         | 1,12,336                      | 93,598.5                                             | 16.67                        | 17.24        | 15.91                         |
>
>
> As we can see, the inference cost reductions are significant even when measured by wallclock time. Importantly, the inference time reduction roughly matches the % reduction in tokens generated, which is the original metric we report. This confirms that empirical results match our more hardware-agnostic measures of inference efficiency.

---

### Note · Authors · 2025-08-13

Our paper shows that across a wide range of tasks, datasets and domains, the internal states of an LM can be used to pre-emptively predict how the LM will "behave" on an input. We show that when used in a conformal prediction framework, this enables provably precise a priori behaviour prediction, enabling us to reduce token generation costs, more efficiently safeguard against jailbreaks and detect errors in model output before they occur.

In the initial reviews, most reviewers agreed that the paper was well-written (DPE7, hGPd, fHv2) and that the idea of a priori behaviour prediction is "compelling" and has "strong practical value" with respect to "critical, real-world problems" (DPE7, hGPd, 74n8, fHv2). While some reviewers felt that our claims were substantiated by "extensive experiments", there were a few concerns:

1. Measurement of the inference efficiency of our system: Some reviewers (DPE7, hGPd) worried that our method of quantifying inference efficiency i.e., % reduction in the number of tokens generated, may not be the fairest way to benchmark inference cost, preferring a more direct wallclock measurement instead. During the rebuttal, we provided the experiments that the reviewers asked for, showing that the wallclock metric corresponds to the metric chosen in our paper, and confirming our significant reduction to the inference cost of the system.
2. LM Baseline: Other reviewers (DPE7, fHv2) shared concerns that there was little differentiation between our method of probing internal layers and our baseline of using LMs to fine-tune on the (input text, output behavior) pairs. We directly addressed this concern in multiple ways, providing both conceptual arguments for their difference and showing that empirically, pretrained LMs are incapable of anticipating another LMs future behavior, regardless of scale.

We believe the primary outstanding concern is that by including a wide range of tasks spanning from instruction following to question answering, our paper lacks focus. Respectfully, we disagree that this is a weakness. The intention of our work is specifically to show that on a diverse set of tasks and a wide array of behaviours, the same practice of probing hidden states using our conformal probes can precisely and preemptively identify behaviour. It is the width and coverage of our tasks that allow us to show this claim robustly, and we believe it is a strength of our work that the same method shows success across a range of settings.

---

### Decision · Program_Chairs · 2025-09-17

**Decision:**

Accept (poster)

**Comment:**

# Summary
This paper introduces a conformal probe method that can predict several types of model behavior from the hidden states of the final input token, before generating any new tokens. The method trains a linear classifier to predict behavior labels from the hidden states, and then calibrates the classifier's prediction using conformal prediction. The calibrated classifier prediction generates an early-warning or early-exit signal to predict failures in safety alignment, instruction following, or CoT reasoning in advance. With high prediction accuracy for certain behaviors, the proposed method can significantly reduce inference costs. The experiments demonstrate the method's effectiveness on 27 datasets.

# Strengths
- The proposed method is simple, effective, and easy to follow.
- The idea of preemptive behavior prediction is innovative.
- The paper showcases several practically important applications of the proposed technique, where it can save the cost of generating many output tokens.
- The experiments provided in the original draft and rebuttal are comprehensive and provide insights into when the conformal probe is effective.

# Weaknesses
- Despite the novelty of preemptive behavior prediction, the application of probing and conformal prediction is shallow and incremental.
- Some reviewers pointed out that the paper lacks focus on the downstream tasks.
- The fairness of the initial efficiency comparison is questionable as it did not include the cost/latency of probe training/inference and report wall-clock time. It also does not cover the cost of collecting the training set for the probe.
- It is not fully clear whether "the model can indeed predict its own behavior", or another irrelevant but similar-sized model can be trained to achieve the same capability.
- Generalization of the conformal probe under domain or distribution shifts needs to be studied.

# Reasons to Accept
- Preemptive prediction of model behaviors from its hidden states is an important and novel problem to study.
- The proposed method is simple to use and straightforward to understand.
- There are various potential applications of the proposed method.
- The rebuttal and discussion addressed the main concerns from the reviewers.

# Summary of Discussion
- In the rebuttal, the authors provided additional experiments on more baselines, reported a more comprehensive analysis of the efficiency, and the results on models from different families.
- In the discussion, all the reviewers responded that their main concerns have been addressed and confirmed their support to the submission.